# ICU-Admission Hyperphosphataemia Is Related to Shock and Tissue Damage, Indicating Injury Severity and Mortality in Polytrauma Patients

**DOI:** 10.3390/diagnostics11091548

**Published:** 2021-08-26

**Authors:** Christopher Rugg, Mirjam Bachler, Robert Kammerlander, Daniel Niederbrunner, Johannes Bösch, Stefan Schmid, Janett Kreutziger, Mathias Ströhle

**Affiliations:** Department of Anaesthesiology and Critical Care Medicine, Medical University of Innsbruck, Anichstrasse 35, 6020 Innsbruck, Austria; mirjam.bachler@tirol-kliniken.at (M.B.); robert.kammerlander@student.i-med.ac.at (R.K.); daniel.niederbrunner@student.i-med.ac.at (D.N.); johannes.boesch@tirol-kliniken.at (J.B.); stefan.schmid@tirol-kliniken.at (S.S.); janett.kreutziger@i-med.ac.at (J.K.); mathias.stroehle@tirol-kliniken.at (M.S.)

**Keywords:** hyperphosphatemia, phosphate, ICU, polytrauma

## Abstract

Hyperphosphataemia can originate from tissue ischaemia and damage and may be associated with injury severity in polytrauma patients. In this retrospective, single-centre study, 166 polytrauma patients (injury severity score (ISS) ≥ 16) primarily requiring intensive care unit (ICU) treatment were analysed within a five-year timeframe. ICU-admission phosphate levels defined a hyperphosphataemic (>1.45 mmol/L; *n* = 56) opposed to a non-hyperphosphataemic group (*n* = 110). In the hyperphosphataemic group, injury severity was increased (ISS median and IQR: 38 (30–44) vs. 26 (22–34); *p* < 0.001), as were signs of shock (lactate, resuscitation requirements), tissue damage (ASAT, ALAT, creatinine) and lastly in-hospital mortality (35.7% vs. 5.5%; *p* < 0.001). Hyperphosphataemia at ICU admission was shown to be a risk factor for mortality (1.46–2.10 mmol/L: odds ratio (OR) 3.96 (95% confidence interval (CI) 1.03–15.16); *p* = 0.045; >2.10 mmol/L: OR 12.81 (CI 3.45–47.48); *p* < 0.001) and admission phosphate levels alone performed as good as injury severity score (ISS) in predicting in-hospital mortality (area under the ROC curve: 0.811 vs. 0.770; *p* = 0.389). Hyperphosphataemia at ICU admission is related to tissue damage and shock and indicates injury severity and subsequent mortality in polytrauma patients. Admission phosphate levels represent an easily feasible yet strong predictor for in-hospital mortality.

## 1. Introduction

In the human body, the vast majority (~85%) of phosphate is stored as crystalline calcium phosphate in the bones. The second largest compound can be found intracellularly (~14%), with intracellular phosphate concentrations commonly described up to 100-fold greater than in plasma [1]. However, focussing on free soluble anions only, studies revealed intracellular phosphate concentrations of 0.78 mmol/L in human erythrocytes, 2.01 mmol/L in human leukocytes, 4.12 mmol/L in human thrombocytes, 10 mmol/L in canine muscle cells and 14 mmol/L in Ehrlich ascites tumour cells from mice [2,3,4]. A far greater extent of intracellular phosphate is present organically bound, notably in energy carriers (e.g., ATP, phosphocreatine, hexose-phosphate), nucleosides in DNA or RNA and other important compounds (e.g., 2,3-bisphosphoglycerate or phospholipids) [5]. Merely approximately 1% of the total body phosphate storage is in solution as anion extracellularly. The complex regulation of phosphate homeostasis involves diverse hormones (parathormone, vitamin D, fibroblast growth factor 23) and is strongly dependent on sufficient renal excretion [6,7]. Basically speaking, hyperphosphataemia occurs due to an imbalance between phosphate load and elimination. In detail, insufficient excretion is often due to acute or chronic kidney disease and an increased load either originates from exogenous administration or endogenously by transcellular shift [8]. Hereby, cellular disintegration (e.g., tumour lysis syndrome, haemolysis, rhabdomyolysis, trauma) but also organ malperfusion and consequent ischaemia can acutely lead to a phosphate accumulation exceeding renal excretion capabilities. Regarding the latter, the pathophysiological source behind this phenomenon includes a strongly increased release of inorganic from formerly organically bound phosphates in the presence of cellular hypoxia (esp. ATP → AMP + 2P_i_) [9,10,11,12,13].

With regard to possible consequences, symptoms of acute hyperphosphataemia are usually mild and mainly limited to those of a concomitantly developing hypocalcaemia (e.g., muscular weakness, tetany, confusion, cardiac arrhythmias, hypotension). Chronically elevated phosphate levels have particularly been associated with cardiovascular risk [14,15] and mortality in chronic kidney disease [16,17,18]. However, also in general hospital populations an increased risk for acute kidney injury [19], respiratory failure [20] and mortality has been shown [21,22]. With regard to specific populations, admission hyperphosphataemia has furthermore been identified as risk factor for mortality in patients suffering from community acquired pneumonia [23] or severe burns [24] or generally admitted to the emergency room [25].

We hypothesize that polytrauma patients are capable of developing acute hyperphosphataemia due to tissue damage and ischaemia, indicating injury severity. Furthermore, we seek to elucidate its role as possible predictor of mortality.

## 2. Materials and Methods

This retrospective study was approved by the Ethics Committee of the Medical University of Innsbruck (20 January 2021; EK Nr.: 1399/2020) and the Institutional Review Board. Due to its retrospective design a consent to participate was not applicable.

The study was conducted at the traumatological intensive care unit (ICU) of the Innsbruck Medical University Hospital. Containing 11 level-3 beds, approximately 300–350 patients per year are treated mainly following multiple trauma as well as scheduled and emergency surgery (cardiac, vascular, thoracic, abdominal, traumatological and transplantations). Polytrauma patients—defined as injury severity score (ISS) ≥ 16—treated between 1 January 2015 and 31 December 2019 were obtained from our documentation database. This primary query resulted in 237 eligible patients (Figure A1). By further analysing the ICU patient data management system and the hospital information system additional data were obtained. Besides age, gender, body mass index (BMI) and pre-existing comorbidities, these data included the injury severity score including corresponding abbreviated injury scales (AIS; Association for the Advancement of Automotive Medicine 2008), simplified acute physiology score 3 (SAPS 3) [26,27] and sequential organ failure assessment (SOFA) score on admission [28,29], body temperature at hospital arrival, existing traumatic brain injury, resuscitation requirements (packed red blood cells, fresh frozen plasma, colloids, crystalloids) as well as additional admission lab values like serum glucose, lactate, phosphate, creatinine, aspartate aminotransferase (ASAT), alanine aminotransferase (ALAT), lactate dehydrogenase (LDH) and myoglobin. In-hospital as well as ICU length of stay (LOS) and mortality were documented. With regard to comorbidities, “other” refers to comorbidities besides those explicitly mentioned (arterial hypertension, cerebrovascular disease, chronic obstructive pulmonary disease, coronary artery disease, diabetes mellitus type 2, heart failure, chronic kidney disease, peripheral artery disease) and “any” classifies a patient suffering from at least one comorbidity.

Patients or the public were not involved in the design, or conduct, or reporting, or dissemination plans of our research.

Data were primarily stored in Excel (v16.4, Microsoft, Redmond, WA, USA). After importing, further statistical analysis was performed utilizing R (v4.0.2, R Core Team, www.R-project.org) and RStudio (v1.2.5001, RStudio, Inc., Boston, MA, USA).

Thirty-one patients were excluded due to incomplete data on ICU admission and 33 because ICU admission did not occur primarily after hospital admission or initial emergency surgery. Another seven were excluded due to pre-existing chronic kidney disease (Figure A1).

On the basis of ICU admission phosphate levels and laboratory defined threshold values, a hyperphosphataemic (>1.45 mmol/L) and a non-hyperphosphataemic (≤1.45 mmol/L) group were determined. Data are presented as count and percentage or due to non-normal distribution as median and interquartile range (IQR) where applicable. Regarding demographics, Chi-squared test was performed to detect group differences in frequencies and Mood’s median test for group differences of continuous data. A Bonferroni-correction for multiple testing (*n* = 40) was performed.

Admission markers of tissue damage (ASAT, ALAT, LDH, myoglobin, creatinine) and ischemia (lactate, glucose, resuscitation requirements) were compared between patients presenting with hyperphosphataemia and those not. Where applicable, the correlation with admission phosphate levels was additionally analyzed via Spearman’s rho.

Considering clinical relevance, mortality analysis was conducted after subdividing patients by their admission phosphate levels in a hypophosphataemic (<0.80 mmol/L), a normophosphataemic (as defined by laboratory thresholds: 0.80–1.45 mmol/L) and two hyperphosphataemic groups (1.46–2.10 mmol/L and >2.10 mmol/L). The threshold of 2.10 mmol/L in the hyperphosphataemic group was defined by creating equal steps in phosphate levels (0.80–1.45–2.10 mmol/L). Mortality rates were calculated for each group and presented graphically via spline transformation. The relation of admission hyperphosphataemia and increase in in-hospital mortality was then quantified by odds ratios (OR) and 95% confidence intervals (CI) in a logistic regression model adjusted for age, sex, pre-existing comorbidities, injury severity and corresponding phosphate levels at ICU admission.

Lastly, receiver operating characteristics (ROC) curve analysis was performed to determine the extent of in-hospital mortality prediction for ICU admission phosphate levels and SAPS 3, SOFA and ISS values. Test quality was quantified by area under the ROC curve (AUC) analysis and compared among each other with a paired DeLong’s test for two correlated ROC curves.

In general, a *p*-value < 0.05 was considered significant.

## 3. Results

A total of 166 polytrauma patients (ISS ≥ 16) requiring ICU treatment in the observed five-year timeframe were analysed (Figure A1). Table 1 depicts general demographics, injury severity and baseline characteristics for all patients as well as in comparison for those presenting with hyperphosphataemia (*n* = 56) at ICU admission and those not (*n* = 110).

While age, gender distribution, BMI and pre-existing comorbidities did not differ between the two groups, hyperphosphataemia at ICU admission was associated with a greater injury severity as shown by the ISS (38 (30–44) vs. 26 (22–34); *p* < 0.001). Initial SAPS 3 (63 (52–71) vs. 44 (35–56); *p* < 0.001) and SOFA-score (12 (10–14) vs. 9 (6–10); *p* < 0.001) were also clearly elevated when compared to non hyperphosphataemic patients. AIS analysis revealed a predominant increase in abdominal injury severity (3 (0–4) vs. 1 (0–2); *p* = 0.002) when presenting with elevated phosphate levels at ICU admission (Table 1).

ICU-admission markers of tissue damage (ASAT, ALAT, creatinine) and shock (lactate, glucose) were clearly increased in the hyperphosphataemic group, as were resuscitation requirements (fresh frozen plasma, norepinephrine) on the first day of admission (Table 2). Admission phosphate levels showed best correlation with admission lactate levels (Spearman’s rho = 0.65; *p* < 0.001) and poorer correlation with markers of tissue damage on admission (Table 2).

ICU- as well as in-hospital-mortality was clearly increased when presenting with hyperphosphataemia at ICU admission (32.1% and 35.7% vs. 4.5% and 5.5%; *p* < 0.001) (Table 1). The admission-phosphate dependent rise in in-hospital mortality is shown in Figure 1.

Multivariate logistic regression analysis adjusting for age, gender, pre-existing comorbidities, ISS and admission phosphate levels revealed age and hyperphosphataemia at ICU admission to be risk factors for in-hospital mortality (Table 3). Although significant in the univariate analysis, ISS missed significance in the multivariate model.

By area under the ROC-curve analysis, the quality of ICU admission phosphate levels in predicting in-hospital mortality was analysed and compared to initial ISS, SOFA- and SAPS 3-scores (Figure 2). Hereby, mere admission phosphate levels (AUC: 0.811) performed as good as ISS (AUC: 0.770; *p* = 0.389), SAPS 3 (AUC: 0.880; *p* = 0.114) and SOFA-scores (AUC: 0.877; *p* = 0.061).

## 4. Discussion

In this retrospective, single-centre study 166 polytrauma patients (ISS ≥ 16) primarily requiring ICU treatment were analysed within a five-year timeframe. Those presenting with hyperphosphataemia at ICU admission had a significantly higher injury severity and increased signs of shock (lactate, glucose, resuscitation requirements) as well as tissue damage (ASAT, ALAT, creatinine) when compared to non-hyperphosphataemic patients. Particularly the severity of abdominal injuries were increased in the hyperphosphataemic group. Admission phosphate levels correlated well with markers of tissue damage (ASAT, ALAT, creatinine) but better with markers of tissue ischaemia (lactate). Hyperphosphataemic patients had a significantly increased in-hospital and ICU mortality. Logistic regression analysis confirmed that admission hyperphosphataemia was correlated with an increased risk of in-hospital mortality. Area under the ROC curve analysis revealed that ICU admission phosphate levels alone performed as good as ISS, SAPS 3 and SOFA-scores in predicting in-hospital mortality.

A bulk of literature concentrates on the origin and the consequences of hyperphosphataemia in chronic kidney and cardiovascular disease [14,15,16,17,18,30,31,32,33] but little is known about cause and effect of elevated phosphate levels beyond these entities. With regard to our cohort of polytrauma patients without pre-existing chronic kidney disease, we assume admission hyperphosphataemia to rather be caused by excessive release than by impaired excretion.

Substantial cellular breakdown with release of greater amounts of intracellular, high phosphate containing fluid is known to cause hyperphosphataemia in entities like tumour lysis syndrome and severe rhabdomyolysis or haemolysis. In patients presenting with hyperphosphataemia in this study, unspecific markers of tissue damage (e.g., ASAT, ALAT) were also clearly elevated, strongly suggesting its contribution to elevated phosphate levels.

Besides actual cellular disintegration, tissue ischaemia as well, has been shown to be able to cause a rise in phosphate levels. As far back as 1966, lactic acidosis was shown to be correlated with hyperphosphataemia by Tranquada et al. [12]. Since then, his results have been confirmed in further studies [9,10] and hyperphosphataemia has also been associated with haemorrhagic shock [13] and diabetic ketoacidosis [11]. An answer to the origin of this phenomenon has been given in an extensive study on the effects of ischaemia on the content of metabolites in rat liver and kidney in vivo [9]. Besides different intermediates of glycolysis, particularly adenosine nucleotides and phosphate levels were measured one, two and five minutes after severance of blood supply and compared to baseline. Hereby, in both tissues inorganic phosphate levels increased rapidly two to three-fold, mainly as the result of the dephosphorylation of adenosine nucleotides. Similar results have also been shown for the ischaemic canine myocardium [34]. Lastly, in the ischaemic tissue, inorganic phosphate is progressively released as ATP is utilized more than it is regenerated. Concomitantly, lactate formation increases in the presence of anaerobic glycolysis.

Tissue ischaemia in polytrauma patients will mainly result from global circulatory shock (especially haemorrhagic) or local perfusion deficits. In the hyperphosphataemic group in this study, surrogates of tissue ischaemia and ongoing shock were clearly elevated on ICU admission. Besides resuscitation requirements including fresh frozen plasma and norepinephrine administration, also lactate and glucose levels were significantly increased. Elevated blood glucose levels assessed early on emergency room admission have been associated with the degree of haemorrhagic shock and also mortality in polytrauma patients [35,36,37]. Hereby, one recognized mechanism influencing this stress hyperglycaemia is developing insulin resistance. Noteworthy, glucose and phosphate are connected via insulin mediated intracellular shift. Whether stress induced insulin resistance also contributes to hyperphosphataemia by hindering intracellular shift or both are fairly independent signs of shock cannot be said in this study neither can the extent of their possible effects.

Particularly abdomen injuries were significantly more severe in patients presenting with hyperphosphataemia. An increased risk of developing (haemorrhagic) shock or organ ischaemia from severe injuries to this area can be suspected. Importantly, elevated tissue damage markers did not only origin from liver injury in the hyperphosphataemic group, as seen by the attenuated increase in ALAT levels especially when compared to ASAT levels.

Furthermore, a correlation between admission phosphate and admission markers of tissue damage and ischaemia was shown. Hereby, the correlation with lactate was stronger than with LDH, ASAT, ALAT, myoglobin or creatinine suggesting tissue ischaemia to contribute stronger to hyperphosphataemia than direct cellular breakdown in our cohort.

Regarding resuscitation management, our department prioritizes a coagulation factor concentrate based strategy [38] as also proposed by the European guideline on the management of major bleeding and coagulopathy following trauma [39]. The resulting increased use of colloids over fresh frozen plasma nearly exclusively consists of succinylated gelatine (Gelofusin^®^, BBraun, Germany).

With regard to mortality, a comparable study on severely burned patients also showed an association of admission hyperphosphataemia with higher mortality [24]. The link to circulatory shock and tissue damage was not discussed, as focus was set on possible kidney injury. Otherwise, large studies on unselected hospital populations showed an increased mortality and rate of renal and respiratory failures when presenting with hyperphosphataemia at hospital admission [19,20,21,22,25]. Risk assessment in these studies on unselected patients was usually adjusted for demographics (age, gender, race) and pre-existing comorbidities, including chronic kidney disease, but not for admission diagnoses. In a study on the effect of admission hyperphosphataemia on developing acute kidney injury it was noted that often patients “with a primary diagnosis of haematology/oncology presented with higher admission serum phosphate levels” [19]. To what extent, hyperphosphataemia actively affects outcome or merely acts as an epiphenomenon of other underlying pathologies remains difficult to discriminate in these unselected patients. It is known that hyperphosphataemia can directly cause acute kidney injury via crystalline tubular damage [40,41] and at least directly impact cardiovascular disease and respiratory failure via vascular medial calcification and muscular weakness [20,33]. In our study, 13 of 56 hyperphosphataemic patients (23.2%) required renal replacement therapy due to acute kidney injury during ICU treatment compared to 3 of 110 patients (2.7%, *p* < 0.001) without hyperphosphataemia on ICU admission. At the same time, injury severity in general, and particularly signs of shock or even serum-myoglobin, were clearly increased in these patients, again complicating cause and effect differentiation.

Admission phosphate levels alone performed as good as ISS in predicting in-hospital mortality. Although hyperphosphataemia itself can contribute to undesirable events, we conclude that in polytrauma patients elevated admission phosphate levels result from tissue damage and ischaemia and rather represent injury severity than a direct cause of mortality. In summary, admission phosphate levels represent an easily feasible and simple marker of tissue ischaemia and damage indicating injury severity and mortality in polytrauma patients.

The main limitation of this study is certainly its small sample size. As one of a few consequences, a very wide 95% confidence interval can be observed within the performed logistic regression analysis. Further limitations are the retrospective study design within one study centre. Bias due to centre specific characteristic cannot be excluded. Larger, prospective and possibly multi-centre studies are required to consolidate our results.

## 5. Conclusions

Hyperphosphataemia at ICU admission relates to shock and tissue damage and indicates injury severity and subsequent mortality in polytrauma patients without pre-existing renal comorbidities. Particularly the severity of abdominal injuries was increased in the hyperphosphataemic group. Mere admission phosphate levels represent an easily feasible yet strong predictor for in-hospital mortality.

## Figures and Tables

**Figure 1 diagnostics-11-01548-f001:**
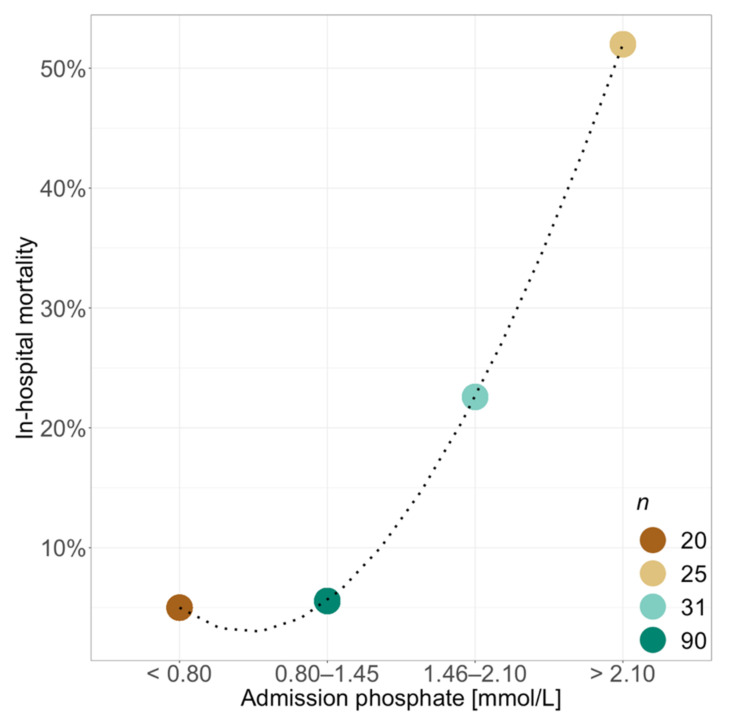
Association between ICU-admission phosphate levels and in-hospital mortality. *n* equals total number of patients per group. Dashed line derived from spline transformation.

**Figure 2 diagnostics-11-01548-f002:**
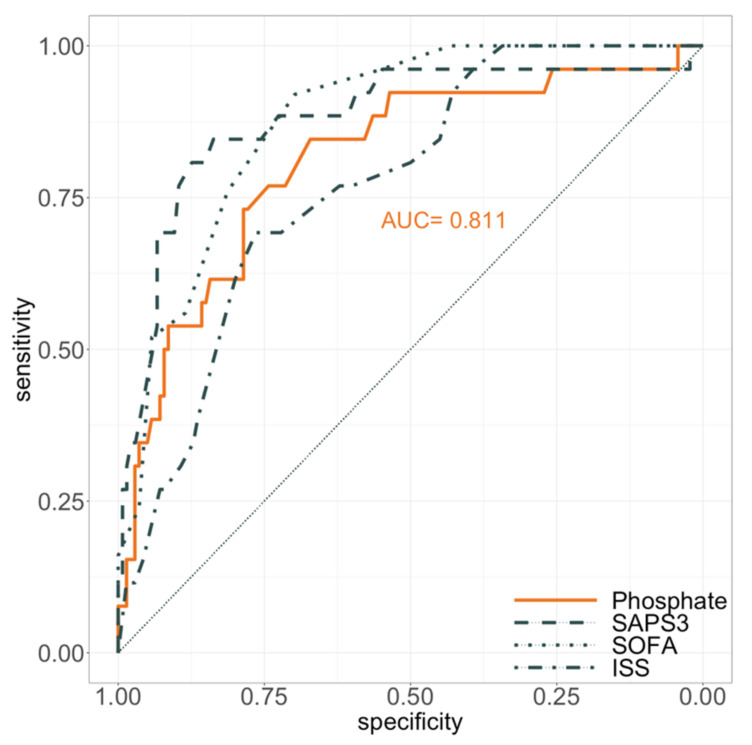
Receiver operating characteristics of ICU admission values regarding in-hospital mortality. SOFA: sequential organ failure assessment; SAPS: simplified acute physiology score; ISS: injury severity score.

**Table 1 diagnostics-11-01548-t001:** General demographics, injury severity and baseline characteristics.

	All Patients	Admission Phosphate	Admission Phosphate	*p*	
		≤1.45 mmol/L	>1.45 mmol/L		*p*
	(*n* = 166)	(*n* = 110)	(*n* = 56)		(Bonferroni Corrected)
	*n* (%) or Median (IQR)	*n* (%) or Median (IQR)	*n* (%) or Median (IQR)		
ICU-admission phosphate [mmol/L]	1.20 (0.92–1.57)	1.00 (0.86–1.20)	1.94 (1.57–2.68)	<0.001	<0.001
Age [yrs]	47 (31–57)	47 (33–57)	46 (28–58)	0.743	1
Sex					
Male	129 (77.7)	90 (81.8)	39 (69.6)	0.113	1
Female	37 (22.3)	20 (18.2)	17 (30.4)		
BMI [kg/m^2^]	24.7 (23.1–26.3)	24.7 (23.4–26.3)	25.4 (22.5–26.3)	0.289	1
Comorbidities					
Any	61 (36.7)	41 (37.3)	20 (35.7)	0.979	1
Arterial hypertension	25 (15.1)	17 (15.5)	8 (14.3)	1.000	1
Cerebrovascular disease	5 (3.0)	5 (4.5)	0 (0)	0.254	1
COPD	3 (1.8)	2 (1.8)	1 (1.8)	1.000	1
Coronary artery disease	11 (6.6)	8 (7.3)	3 (5.4)	0.889	1
Diabetes mellitus type 2	5 (3.0)	4 (3.6)	1 (1.8)	0.858	1
Heart failure	7 (4.2)	4 (3.6)	3 (5.4)	0.910	1
Chronic kidney disease	0 (0)	0 (0)	0 (0)	1.000	1
Peripheral artery disease	0 (0)	0 (0)	0 (0)	1.000	1
Other	50 (30.1)	35 (31.8)	15 (26.8)	0.625	1
Injury severity score	29 (22–41)	26 (22–34)	38 (30–44)	<0.001	<0.001
Abbreviated injury scale					
Head or neck	2 (0–4)	2 (0–4)	2 (0–4)	0.576	1
Face	0 (0–1)	0 (0–1)	0 (0–0)	0.141	1
Chest	3 (3–4)	3 (2–4)	3 (3–4)	0.081	1
Abdomen	2 (0–3)	1 (0–2)	3 (0–4)	<0.001	0.002
Extremities/pelvic girdle	2 (0–3)	2 (0–3)	2 (0–3)	0.796	1
External	0 (0–1)	0 (0–1)	0 (0–1)	0.447	1
Body temp. at hospital arrival [°C]	35.4 (34.8–36.4)	36.0 (35.3–36.5)	35.0 (33.4–35.5)	0.013	0.521
SAPS 3 (ICU admission)	50 (38–63)	44 (35–56)	63 (52–71)	<0.001	<0.001
SOFA Score (ICU admission)	10 (7–12)	9 (6–10)	12 (10–14)	<0.001	<0.001
Intensive care unit					
Length of stay [d]	7 (3–14)	8 (2–14)	7 (3–12)	0.828	1
Mortality	23 (13.9)	5 (4.5)	18 (32.1)	<0.001	<0.001
In-hospital					
Length of stay [d]	17 (9–34)	21 (11–34)	12 (4–27)	0.001	0.043
Mortality	26 (15.7)	6 (5.5)	20 (35.7)	<0.001	<0.001

BMI: body mass index; COPD: chronic obstructive pulmonary disease; SOFA: sequential organ failure assessment; SAPS: simplified acute physiology score; ISS: injury severity score; ICU: intensive care unit.

**Table 2 diagnostics-11-01548-t002:** ICU-admission phosphate in correlation to markers of tissue damage, shock and resuscitation efforts.

	Admission Phosphate	Admission Phosphate	*p*		Correlation with Admission Phosphate Levels Spearman’s Rho (*p*)
	≤1.45 mmol/L	>1.45 mmol/L		*p*	
	(*n* = 110)	(*n* = 56)		(Bonferroni Corrected)	
	*n* (%) or Median (IQR)	*n* (%) or Median (IQR)			
Markers of tissue damage on admission					
ASAT [U/L]	72 (46–154)	170 (90–431)	0.001	0.043	0.44 (<0.001)
ALAT [U/L]	53 (33–121)	120 (62–319)	0.001	0.043	0.39 (<0.001)
LDH [U/L]	341 (253–425)	485 (317–666)	0.005	0.187	0.48 (<0.001)
Myoglobin [µg/L]	1840 (518–2094)	3037 (1454–6362)	0.002	0.078	0.44 (<0.001)
Creatinine [mg/dL]	0.96 (0.86–1.10)	1.25 (1.08–1.47)	<0.001	<0.001	0.49 (<0.001)
Markers of shock on admission					
Lactate [mg/dL]	16 (11–23)	46 (25–78)	<0.001	<0.001	0.65 (<0.001)
Glucose [mg/dL]	138 (116–165)	176 (137–255)	<0.001	0.029	0.39 (<0.001)
Admission day resuscitation ^#^					
Packed red blood cells [mL]	0 (0–560)	500 (0–1585)	0.029	1	
Fresh frozen plasma [mL]	0 (0–0)	0 (0–288)	<0.001	0.024	
Colloids [mL]	3002 (2000–4500)	4002 (2500–5500)	0.014	0.575	
Crystalloids [mL]	2449 (1601–3497)	2188 (1569–3516)	0.326	1	
Norepinephrine [µg/kg/min] *	0.11 (0.04–0.21)	0.28 (0.16–0.39)	<0.001	0.001	

^#^ Including all fluids given pre- and post-ICU-admission on day of admission; * presented as maximum requirement on admission day.

**Table 3 diagnostics-11-01548-t003:** Logistic regression analysis. Odds ratios for in-hospital mortality in polytrauma patients.

Variable	Crude Odds Ratio	Adjusted Odds Ratio	*p*
	(95% CI)	(95% CI)	
Age	1.03 (1.00–1.05)	1.03 (1.00–1.07)	0.048
Sex (female vs. male)	1.70 (0.67–4.30)	1.35 (0.41–4.53)	0.622
Pre-existing comorbidity	1.09 (0.46–2.58)	0.93 (0.30–2.88)	0.896
Injury severity score	1.07 (1.03–1.10)	1.03 (1.00–1.07)	0.089
Admission phosphate levels			
0.80–1.45 mmol/L	reference		
<0.80 mmol/L	0.89 (0.10–8.11)	0.95 (0.10–8.99)	0.966
1.46–2.10 mmol/L	4.96 (1.44–17.03)	3.96 (1.03–15.16)	0.045
>2.10 mmol/L	18.42 (5.57–60.87)	12.81 (3.45–47.48)	<0.001

## Data Availability

The datasets used and/or analysed during the current study are available from the corresponding author on reasonable request.

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
