# Peer review of "ICU-Admission Hyperphosphataemia Is Related to Shock and Tissue Damage, Indicating Injury Severity and Mortality in Polytrauma Patients"

_diagnostics, 2021, doi:10.3390/diagnostics11091548_

Round 1

Reviewer 1 Report

Regarding the sentence 191, the author mentioned "Logistic regression analysis confirmed that admission hyperphosphataemia independently increased the risk of inhospital mortality.". However, in this study, the author can only demonstrate correlation but not causation. Therefore, the author can not tell hyperphosphataemia independently increased the risk of inhospital mortality. Of note, the author seems aware of the difference between correlation and causation, saying "Although hyperphosphataemia itself can contribute to undesirable events, we conclude that in polytrauma patients elevated admission phosphate levels result from tissue damage and ischaemia and rather represent injury severity than a direct cause of mortality." in 273-275. These inconsistent statements may confuse readers. 

Regarding 242-244, the author said "the correlation with lactate was stronger than with LDH, ASAT, ALAT, myoglobin or creatinine suggesting tissue ischaemia to contribute stronger to hyperphosphataemia than direct cellular breakdown in our cohort. It is unclear to me whether tissue ischemia can cause hyperphosphatemia without cellular/tissue breakdown or other cause of hyperphosphatemia. 

Author Response

Dear Reviewer,

We honestly thank you for your time as well as your favorable review and valuable comments. In the following we hope to sufficiently address the issues raised by you.

Regarding the sentence 191, the author mentioned "Logistic regression analysis confirmed that admission hyperphosphataemia independently increased the risk of inhospital mortality.". However, in this study, the author can only demonstrate correlation but not causation. Therefore, the author can not tell hyperphosphataemia independently increased the risk of inhospital mortality. Of note, the author seems aware of the difference between correlation and causation, saying "Although hyperphosphataemia itself can contribute to undesirable events, we conclude that in polytrauma patients elevated admission phosphate levels result from tissue damage and ischaemia and rather represent injury severity than a direct cause of mortality." in 273-275. These inconsistent statements may confuse readers. 

We must apologize for this confusion!

We completely agree and adapted the mentioned sentence:

Page 7 line 269ff: “Logistic regression analysis confirmed that admission hyperphosphataemia was correlated with an increased risk of in-hospital mortality”

Regarding 242-244, the author said "the correlation with lactate was stronger than with LDH, ASAT, ALAT, myoglobin or creatinine suggesting tissue ischaemia to contribute stronger to hyperphosphataemia than direct cellular breakdown in our cohort. It is unclear to me whether tissue ischemia can cause hyperphosphatemia without cellular/tissue breakdown or other cause of hyperphosphatemia. 

Dear Reviewer,

We do understand your concerns and would like to share our thoughts on this topic.

The theory behind this rise in phosphate levels is probably best described as opposite of what happens during the Refeeding-Syndrome where an increased intracellular shift leads to a strong decrease in serum phosphate levels. In our case we assume an increased extracellular shift, without actual cellular breakdown.

As shown by Tranquada et al. [12] lactic acidosis has been known to be correlated with hyperphosphatemia since the 1960s. As lactate was not routinely measured back then, the combination of an increased anion gap metabolic acidosis with a concomitant hyperphosphatemia was used as surrogate for a lactic acidosis. (O’Connor in 1977 [10 ] )

Hems et al 1970 [8], showed that during tissue ischemia, phosphate levels rise significantly due to a degradation of ATP and ADP to AMP and phosphorous. This phosphorous is than released into circulation by (of course) cellular breakdown but also by transcellular shift due to a highly elevated intra- to extracellular gradient.

Exact mechanisms may not be known, but, provocatively speaking, lactate and many other by-products of cellular metabolism are also released into circulation when intracellularly elevated without cellular breakdown.

The section on this topic within the discussion is just before the cited sentence by you and reads as follows:

“Besides actual cellular disintegration, tissue ischaemia as well, has been shown to be able to cause a rise in phosphate levels. As far back as 1966, lactic acidosis was shown to be correlated with hyperphosphataemia by Tranquada et al. [12]. Since then, his results have been confirmed in further studies [9,10] and hyperphosphataemia has also been associated with haemorrhagic shock [13] and diabetic ketoacidosis [11]. An answer to the origin of this phenomenon has been given in an extensive study on the effects of is-chaemia on the content of metabolites in rat liver and kidney in vivo [9]. Besides different intermediates of glycolysis, particularly adenosine nucleotides and phosphate levels were measured one, two and five minutes after severance of blood supply and compared to baseline. Hereby, in both tissues inorganic phosphate levels increased rapidly 2 to 3-fold, mainly as the result of the dephosphorylation of adenosine nucleotides. Similar results have also been shown for the ischaemic canine myocardium [34]. Lastly, in the ischaemic tissue, inorganic phosphate is progressively released as ATP is utilized more than it is regenerated. Concomitantly, lactate formation increases in the presence of anaerobic glycolysis.

Tissue ischaemia in polytrauma patients will mainly result from global circulatory shock (especially haemorrhagic) or local perfusion deficits. In the hyperphosphataemic group in this study, surrogates of tissue ischaemia and ongoing shock were clearly el-evated on ICU admission. Besides resuscitation requirements including fresh frozen plasma and norepinephrine administration, also lactate and glucose levels were signif-icantly increased.”

Reviewer 2 Report

I would like to thank the authors for their work.  I was invited to review the statistical side of the manuscript so I would like to make few comments:

  • A single centre and N=166, is a rather small setting and the results are would be taken with caution (the authors recognised it in their limitations section).  I would have liked to see the authors suggesting  of a multi centre study in the future to consolidate their results. 
  • Line 103: The Mann-Whitney U test does not necessarily compare medians, this is true only when the distribution has the same shape in both groups and differs only by its location.  So as the authors use R software,  the library (coin) contains the test on median                     (see https://www.statology.org/moods-median-test-r/ )
  • Tables 1 and 2. There are a 34 significance tests in these tables without any correction for multiple testing. Please consider adjusting with p values or significance level for these comparisons (even if it doesn’t change which are significant). 
  • Table 1.  How different is " Any " from "Other " for comorbidities?
  • Table 2:  The Abbreviated Injury Scale (AIS)  is between 1 and 6,  what does the value "0" represent? if it represents the patients with not having the injury, the median calculation should only include those with an AIS >0.
  • In Table 3 the 95% CI of the odds ratio of the exposure variable "Phosphate levels" are too wide and this is a consequence of the small sample size.

Author Response

I would like to thank the authors for their work.  I was invited to review the statistical side of the manuscript so I would like to make few comments:

Dear Reviewer,

we must dearly thank you for your time and efforts put in our manuscript. We feel very fortunate in being reviewed by someone with statistical expertise and gladly adapted the manuscript upon your comments.

  • A single centre and N=166, is a rather small setting and the results are would be taken with caution (the authors recognised it in their limitations section).  I would have liked to see the authors suggesting  of a multi centre study in the future to consolidate their results. 

We absolutely agree! The single centre design and the small sample size are certainly the main limitations to our study. In order to stress this fact some more we re-wrote the limitations section. We added a sentence regarding the wide confidence interval within our logistic regression analysis and also suggested further larger studies to confirm our results.

The limitation section now reads as follows:

Page 8 line 357: “Main limitation of this study is certainly its small sample size. As one of a few consequences, a very wide 95% confidence interval can be observed within the performed logistic regression analysis. Further limitations are the retrospective study design within one study centre. Bias due to centre specific characteristic cannot be excluded. Larger, prospective and possibly multi-centre studies are required to consolidate our results.“

  • Line 103: The Mann-Whitney U test does not necessarily compare medians, this is true only when the distribution has the same shape in both groups and differs only by its location.  So as the authors use R software,  the library (coin) contains the test on median                     (see https://www.statology.org/moods-median-test-r/ )

  • Tables 1 and 2. There are a 34 significance tests in these tables without any correction for multiple testing. Please consider adjusting with p values or significance level for these comparisons (even if it doesn’t change which are significant). 

Thank you once more for your valuable hints! We gladly visited the stated link and learned about Moods Median Test. We re-did the statistical analysis of the demographics and general characteristic where applicable and also added a Bonferroni-correction.

The p-values in Tables 1 and 2 and also within the running text were modified where necessary. As group differences of some - not so important - values lost significance (body temp, chest AIS, LDH, myoglobin, packed red blood cell and colloid requirement), some small corrections to the results and discussion section were also required. The main results all remained the same, so that modification was truly minor.

In the method section the following was changed:

Page 3 line 102f: “Regarding demographics, Chi-squared test was performed to detect group differences in frequencies and Mood’s Median Test for group differences of continuous data”

            In the results section:

Page 4 line 147f: “AIS analysis revealed a predominant increase in abdominal injury severity (3 (0 – 4) vs. 1 (0 – 2); p= 0.002) when presenting with elevated phosphate levels at ICU admission (Table 1).” (chest injuries were removed from the sentence)

Page 4 line 150ff: “ICU-admission markers of tissue damage (ASAT, ALAT, creatinine) and shock (lactate, glucose) were clearly increased in the hyperphosphataemic group, as were re-suscitation requirements (fresh frozen plasma, norepinephrine) on the first day of ad-mission (Table 2).” (myoglobine, LDH, packed red blood cells and colloids were removed).

Same in the discussion section:

Page 6 line 252ff: “Those presenting with hyperphosphataemia at ICU admission had a significantly higher injury severity and increased signs of shock (lactate, glucose, resuscitation requirements) as well as tissue damage (ASAT, ALAT, creatinine) when compared to non-hyperphosphataemic patients. Particularly the severity of abdominal injuries were increased in the hyperphosphataemic group.”

Page 7 line 276ff: “In patients presenting with hyperphosphataemia in this study, unspecific markers of tissue damage (e.g., ASAT, ALAT) were also clearly elevated, strongly suggesting its contribution to elevated phosphate levels.”

Page 7 line 306ff: “Particularly abdomen injuries were significantly more severe in patients presenting with hyperphosphataemia. An increased risk of developing (haemorrhagic) shock or organ ischaemia from severe injuries to this area can be suspected.”

  • Table 1.  How different is " Any " from "Other " for comorbidities?

We are sorry for this confusion!

In this case, “any” refers to a patient suffering from any kind of comorbidiy (no matter what) and summarizes if the patient was free from any comorbidity or not.

“Other” refers to a comorbidity other than those mentioned in the table. (e.g., psychiatric disease).

We added a sentence to the method section to make this more clearer for the reader:

Page 2 line 88ff: “With regard to comorbidities, other refers to comorbidities besides those explicitly mentioned (arterial hypertension, cerebrovascular disease, chronic obstructive pulmonary disease, coronary artery disease, diabetes mellitus type 2, heart failure, chronic kidney disease, peripheral artery disease) and any classifies a patient suffering from at least one comorbidity.”

  • Table 2:  The Abbreviated Injury Scale (AIS)  is between 1 and 6,  what does the value "0" represent? if it represents the patients with not having the injury, the median calculation should only include those with an AIS >0.

In this point we completely understand your approach. Nevertheless, we do not completely agree. The Injury Severity Score (ISS) is calculated upon the Abbreviated Injury Scales of the three highest injured regions:

 ISS = AISa2 + AISb2 + AISc2

An AIS of 6 in any region equates to death.

Therefore, the highest AIS put into this equation is 5, resulting in a maximum ISS of 75. (If any AIS is 6 the ISS is automatically 75).

Main focus when comparing polytrauma patients is the ISS. The presentation of all AIS scores itemized in a table is not as common. The reason we did it anyway was to analyse and present where these patients were injured (and how bad). In order to properly and also comprehensively present how bad patients were injured where, the 0 indicating uninjured in this region must remain, otherwise the painted picture is skewed toward the severity of those injured in that area only.

One extreme example might best explain our approach:

Lets say we have a large population of two groups allocated by a lab value. This lab value happens to predict brain injury with high sensitivity and specifity (not priorly known). All patients in the group with an elevated lab value have head injuries of medium to severe severity. In the other group however, only a few patients suffer from head injuries. Depending on how much these patients are outliers or not, if I compare the injury severity of only those really suffering from head injuries there might not even be a difference between the groups.

What I want to see however is that the median injury severity of head injuries in the second group is 0 or close to it, as most patient did not suffer any head injury at all.

In our case we want to keep the 0 within the AIS as referral to a patient being uninjured in that body area. The median AISs, then represent the median patient with his median injury pattern (uninjured to the face, slightly injured extremities, and so on)

  • In Table 3 the 95% CI of the odds ratio of the exposure variable "Phosphate levels" are too wide and this is a consequence of the small sample size.

As mentioned above, we added a sentence to the limitations section with regard to the very wide confidence interval in the eye of our small sample size.